# Cost-effectiveness analysis of bundled innovative devices versus standard approach in the prevention of unscheduled peripheral venous catheters removal due to complications in France

Franck Maunoury[1,2]☯*, Bertrand Drugeon[3]☯, Matthieu Boisson[4,5], Nicolas Marjanovic[3], Raphael Couvreur[3,4], Olivier Mimoz[3,4,6], Jeremy Guenezan[3,4,6], on behalf of the CLEAN-3 Study¶

1 Statesia, Le Mans, France, 2 University of Le Mans, GAINS, IRA, Le Mans, France, 3 Emergency Department and Pre-Hospital Care, University Hospital of Poitiers, Poitiers, France, 4 School of Medicine and Pharmacy, University of Poitiers, Poitiers, France, 5 Intensive Care and Peri-Operative Medicine Department, University Hospital of Poitiers, Anesthesia, Poitiers, France, 6 INSERM U1070, Pharmacology of anti-infective drugs, Poitiers, France

☯ These authors contributed equally to this work.
¶ Membership of the CLEAN-3 Study is provided in the Acknowledgments.
* franck.maunoury@statesia.com

## Abstract

The objective of the study was to perform a cost-effectiveness analysis of bundled devices (BDs) versus standard devices (SDs) for the prevention of unscheduled peripheral venous catheter (PVC) removal due to complication from a French investigator-initiated, open-label, single center, randomized-controlled, two-by-two factorial trial (CLEAN-3 study). A 14-day time non homogeneous semi-markovian model was performed to be fitted to longitudinal individual patient data from CLEAN-3 database. This model includes five health states and eight transitional events; a base case scenario, two scenario analyses and bootstrap sensitivity analyses were performed. The cost-effectiveness criterion was the cost per patient with unscheduled PVC removal avoided. 989 adult (age≥18 years) patients were analyzed to compare the BDs group (494 patients), and the SDs group (495 patients). The assessed intervention was a combination of closed integrated catheters, positive displacement needleless-connectors, disinfecting caps, and single-use prefilled flush syringes compared with the use of open catheters and three-way stopcocks for treatment administration. For the base case scenario, an unscheduled 1st PVC removal before discharge was significantly more frequent in the SDs group (235 patients (47.5%) in the SDs group and 172 patients (34.8%) in the BDs group, p = 0.00006). After adjustment for 1st catheter time, the number of patients with unscheduled PVC removal per day was of 16 (95%CI: 15; 18) patients (out of 100) in the BDs group and of 26 (95%CI: 24; 28) patients (out of 100) in the SDs group. The mean cost per patient (adjusted on catheter-time) was of €144 (95%CI: €135-€154) for patients in the SDs group versus €102 (95%CI: €95-€109) for patients in the BDs group; the

**Data Availability Statement:** All relevant data are within the paper and its Supporting Information files.

**Funding:** This study was funded by Becton Dickinson France, https://www.bd.com/fr-fr. University Hospital of Poitiers (OM) received this funding. FM is the CEO of Statesia. A service agreement (contract) was arranged between The University Hospital of Poitiers and Statesia. The assessed bundle of innovative devices is a product marketed by Becton Dickinson. There are no further patents, products in development or marketed products to declare. This does not alter our adherence to all the PLOS ONE policies on sharing data and materials, as detailed online in the guide for authors.The funders had no role in study design, data collection and analysis, decision to publish, or preparation of the manuscript. Statesia provided support in the form of salaries for author FM but did not have any additional role in the study design, data collection and analysis, decision to publish, or preparation of the manuscript. The specific roles of the authors are articulated in the 'author contributions' section.

**Competing interests:** OM & JG received personal fees from Becton Dickinson, funding for congress attendance, and research funding from Becton Dickinson. MB received personal fees from Becton Dickinson. All other investigators and authors declare no competing interests. This does not alter our adherence to PLOS ONE policies on sharing data and materials.

mean saving per patient was of €42 (95%CI: €32-€54). As a consequence, the assessed BDs strategy was less costly and more effective than the SDs strategy.

**Trail registration:** CLEAN-3 study is registered with ClinicalTrials.gov, NCT03757143.

## Introduction

Short-term peripheral venous catheters (PVCs) are the most commonly used invasive medical devices in hospitals, with about 2 billion sold annually worldwide [1]. Unfortunately, peripheral venous catheters (PVC) often fail before the end of treatment due to complications occurrence, including catheter occlusion, skin infiltration (diffusion), phlebitis, catheter dislodgement, and bloodstream or local infections [2]. These complications lead to interruption of treatment, which can be detrimental to patients. Catheter replacement causes pain and induces additional costs [3]. Additionally, bloodstream infections prolong hospitalization and increase treatment costs and mortality [4].

CLEAN-3 clinical study [5], an investigator-initiated, open-label, single center, randomized, two-by-two factorial, superiority trial was performed in 2019 at Poitiers University Hospital, France. Adult patients, visiting the emergency department (ED), with an indication for hospitalization in a medical ward and a PVC for a predicted duration of at least 48 hours were included in the study. Patients were monitored daily for catheter-related complications for up to 48 hours after catheter removal, or earlier if discharged from hospital.

The CLEAN-3 study compares two different approaches to peripheral vascular access. The first one consisted of a bundle of new devices including closed integrated catheters, positive displacement needleless-connectors, disinfecting caps, and single-use prefilled flush syringes. The second was simply a conventional peripheral venous catheter to which the infusion line with a three-way stop cock was directly connected.

CLEAN-3 study [5] showed that the use of the bundle, versus a standard approach, made it possible to (1) reduce the frequency of occurrence of a complication requiring catheter replacement (phlebitis, diffusion, occlusion, local infection, dislodgement) and (2) delay the time of occurrence of these complications. The material of bundle strategy is slightly more expensive (for insertion and per day of use). The standard strategy requires more frequent and earlier removal of a defective catheter and its replacement, which generates an additional expense. Treatment of complications leading to catheter replacement have their own additional costs. The question is whether the extra cost generated by the bundle (at catheter insertion, at each daily use of the venous line and at catheter removal) is offset by the savings generated by (1) less frequent and delayed replacement of the venous line, and (2) less frequent treatment of complications leading to catheter removal.

CLEAN-3 clinical study [5] also confirmed the superiority of 2% chlorhexidine (CHG)– 70% isopropanol over 5% povidone iodine (PVI)– 69% ethanol in reducing both catheter colonization and local infection. This result suggests the use of 2% CHG plus alcohol as the preferred antiseptic for short-term PVC insertion and care. The results for comparison between the two antiseptics were not affected by the type of devices, nor were the results for comparison between standard and innovative devices affected by the type of antiseptic. Moreover, CLEAN-3 clinical study showed that length of stay in hospital were not affected by the choice of antiseptic agent or type of device. As a consequence, two devices groups (combining CHG and PVI solutions) were considered in the cost-effectiveness study: a bundle of innovative devices (BDs), and standard devices (SDs).

The objective of the present study was to perform a cost-effectiveness analysis (CEA) of these devices strategies in the prevention of PVC unscheduled removal due to complications

in France, from modeling techniques based on the CLEAN-3 database. To support the choice of the best devices strategy from a conventional hospital medical care perspective, a decision-analytic model was performed.

## Methods

### Study design

Statistical analysis of observed data from CLEAN-3 database was achieved. The adopted modeling approach complies with the guidelines of French National Authority for Health (Haute Autorité de Santé –HAS) [6]. The 14-day ICU-time non-homogeneous semi-Markovian model structure was based on observed data from the CLEAN-3 study. Modeling and data analyses were performed using Rstudio software (version 1.1.453 – © 2009–2018 RStudio, Inc.). RStudio is an Integrated Development Environment for R (R Core Team (2022). R: A language and environment for statistical computing. R Foundation for Statistical Computing, Vienna, Austria. URL https://www.R-project.org/.).

### Data collection

Data of the cost-effectiveness study were from the CLEAN-3 [5] database delivered by the University Hospital of Poitiers (France). Patients were recruited at the ED before being admitted to the medical wards. The main objective of the clinical study was to hypothesize that skin preparation with 2% chlorhexidine plus 70% isopropanol (chlorhexidine plus alcohol) was more effective than a skin preparation with 5% povidone iodine plus 69% ethanol (povidone iodine plus alcohol) in preventing PVC-related infectious complications. The second assumption testing was to consider that use of closed integrated catheters, positive displacement needleless-connectors, disinfecting caps, and single-use prefilled flush syringes in combination extended the time elapsed between catheter placement and catheter failure compared with the use of open catheters and three-way stopcocks for treatment administration.

### Study population

During the CLEAN-3 study, consecutive adults (age ≥18 years) visiting the ED of the Poitiers University Hospital and requiring a PVC for a predictable duration of at least 48 hours before being admitted to medical wards were enrolled.

The investigators obtained written informed consent before study inclusion from competent patients and at competence recovery from incompetent patients, according to French law. The study was approved by the French Southwest and Overseas Ethics Committee and the French Drug Safety Agency.

The study population included 989 patients after exclusion of 6 patients with failed catheter placement and 5 with withdrawal of consent; 494 patients were randomly assigned to the bundle of innovative devices (BDs) group and 495 patients to the standard devices (SDs) group. Overall, 407 patients (41%) had unscheduled catheter removal for "dislodgement", "diffusion", "local infection", "occlusion" and "phlebitis" complications including 195 men (48%), and 101 patients (25%) had no previous history. Their mean age was 76 years. Two of them died (0.5%) during the study period. 235 patients (47.5%) were from the SDs group and 172 (34.8%) from the BDs group. With respect to qualitative variables, patients in the BDs group were more likely to have past history of heart failure (p = 0.02580) or to be receiving anti-coagulant therapy (p = 0.01858) than patients of the SDs group. For all quantitative variables tested, there was no difference between the BDs and the SDs groups. The duration of hospitalization was

comparable between the BDs group and the SDs group, with durations of 10.4 and 9.3 days, respectively (p = 0.19020).

For comparison, the same statistics were calculated for the population of patients without unscheduled PVC removal (n = 582). Reasons for PVC removal for this group were PVC not useful, scheduled PVC change, suspected infection without identification of an actual complication, or death. Among these patients, with respect to the qualitative variables, there were differences between the two study groups for the absence of past history (34% in the BDs group vs. 44% in the SDs group; p = 0. 01688), history of COPD (13% in the BDs group vs. 6% in the SDs group; p = 0.00779), history of diabetes (20% in the BDs group vs. 14% in the SDs group; p = 0.03765) and history of chronic renal failure (7.5% in the BDs group vs. 3.5% in the SDs group; p = 0.04672). Patients in the BDs group thus appear to be more severe on these variables than those in the SDs group. For all quantitative variables tested, there was no difference between the two study groups except for average age significantly higher in the BDs group (70 years) than in the SDs group (66 years); the catheterization and hospitalization times were not different between the two groups.

## Endpoints

The final health outcome of the cost-effectiveness analysis, adjusted on the 1st catheter-time, was the number of patients with unscheduled PVC removal avoided (per 100 patients) and the cost-effectiveness criterion was the cost per patient with unscheduled PVC removal avoided resulting from BDs devices use, instead of SDs devices.

## Modeling and statistical analysis

Markov models simulate the trajectory of patients among distinct health states over time [7–9]. The main assumption of state-transition Markov models is that the next health state depends only on the present state and not on the sequence of events that preceded it. For an expected goodness of fit to CLEAN-3 data, a multi-state semi-markovian model in continuous time was performed. This type of modeling can take into account iterative occurrences for each health state (e.g., Markov state). This modeling is very close to the daily realities that can be observed through the hospital stay (changes in daily health status, costs of care, etc.). Within this model, transition probabilities between states are time dependent and well suited to individual patient data (IPD) from CLEAN-3 database. This type of modeling is suited to the context of hospital settings where progression of the patient cannot be considered as a long-term condition.

Five health states (Markov states) and 8 transitional events were considered (Table 1) in our cost-effectiveness model. Transitional event is defined as an event that occurs during a cycle and which generates a transition from one model status to another. The follow-up of the patients was related to the first catheter (2 days after its removal, except in the case of death or discharge from hospital in the meantime). In case of a complication justifying the replacement of the catheter, the same device (2nd PVC) was put in place but the follow-up of the patient was stopped, the instructions being to follow the same protocol. The fate of the 2nd catheter (and any subsequent ones) was therefore unknown. The hospital length of stay (LOS) limited to 28 days was recorded.

The type of modeling is a multi-state semi-markovian model in continuous time (transition probabilities are time dependent and fitted to observational IPD from CLEAN-3 database suited to the hospital perspective). For base case scenario, time horizon was of 0–14 days, e.g., the maximum observed duration of catheterization (hospital length of stay: 0–28 days). The Markov Cycle length was the estimated mean sojourn time in each transient health state, for a

**Table 1. Health states defined from the CLEAN-3 clinical study.**

| Health States / Events | Definition |
|---|---|
| Markov state 1: No Event / 1st PVC | Insertion of a first catheter, no event diagnosed |
| Transitional event 1.1: Scheduled PVC removal / No PVC new | Scheduled removal of the 1st catheter / No insertion of a 2nd catheter |
| Transitional event 1.2: Useless PVC / No PVC new | 1st catheter removal because of its useless / No insertion of a 2nd catheter |
| Transitional event 1.3: PVC with suspected infection / No PVC new | 1st catheter removal because of suspected infection / No insertion of a 2nd catheter |
| Markov state 2: No Event / No PVC | No catheter in place, no event diagnosed |
| Transitional event 1.4: PVC dislodgement / PVC new | Unscheduled removal of the 1st catheter due to dislodgement / Insertion of a 2nd catheter |
| Transitional event 1.5: PVC with phlebitis / PVC new | Unscheduled removal of the 1st catheter due to phlebitis / Insertion of a 2nd catheter |
| Transitional event 1.6: PVC with diffusion / PVC new | Unscheduled removal of the 1st catheter due to diffusion / Insertion of a 2nd catheter |
| Transitional event 1.7: PVC with local infection / PVC new | Unscheduled removal of the 1st catheter due to local infection / Insertion of a 2nd catheter |
| Transitional event 1.8: PVC with occlusion / PVC new | Unscheduled removal of the 1st catheter due to occlusion / Insertion of a 2nd catheter |
| Markov state 3: No Event / 2nd PVC | 2nd catheter in place, no event diagnosed |
| Markov state 4: Discharge | Patient leaves the hospital alive |
| Markov state 5: Death | Patient dies during the hospital stay |

PVC: Peripheral Venous Catheter.

given set of covariate values (from observational data in CLEAN-3 database). The number of Markov cycles depends on time horizon. The time horizon was based on the maximum catheter duration observed in each group (SDs group: 336 hours, e.g., 14 days; BDs group: 216 hours, e.g., 9 days) for accounting all types of patients (alive, discharged, or dead). As time horizon observed in CLEAN-3 database was not identical for each group, and for getting a common comparative basis for suited cost-effectiveness analyses, we set the time horizon to 14 days (for base case analysis; all events and catheters were considered from CLEAN-3 database). A scenario analysis considered a time horizon of 2–14 days (subgroup analysis considering patients with a catheter duration of more than 24 hours, or less than 24 hours but with unscheduled catheter removal due to a complication).

The statistical unit of the study is the hospitalized patient within a time horizon of 14 days (including patients discharged alive from the hospital, alive but still in the hospital, or deceased during the hospital stay). Data was censored beyond 14 days as it corresponds to the maximum duration of catheterization observed in CLEAN-3 study. The estimated transition probability matrix is based on individual patient data from CLEAN-3 database. This analysis can be considered as a non-homogeneous Semi-Markov Chain (NH-SMC) analysis which takes into account time dependency of state transition, duration in each health state, and individual path of states through time. The observed transitions among health states are shown on the Markov diagram (Fig 1). In order to be as close as possible to observable realities, we have presented the cost-effectiveness results from the matrix of transition probabilities observed by the model (e.g., 'prevalence.msm' algorithm [10]), and not from the matrix of transition probabilities estimated by the model. This approach is taken by the function prevalence.msm, which constructs a table of observed and expected numbers and percentages of individuals in each state at a set of times.

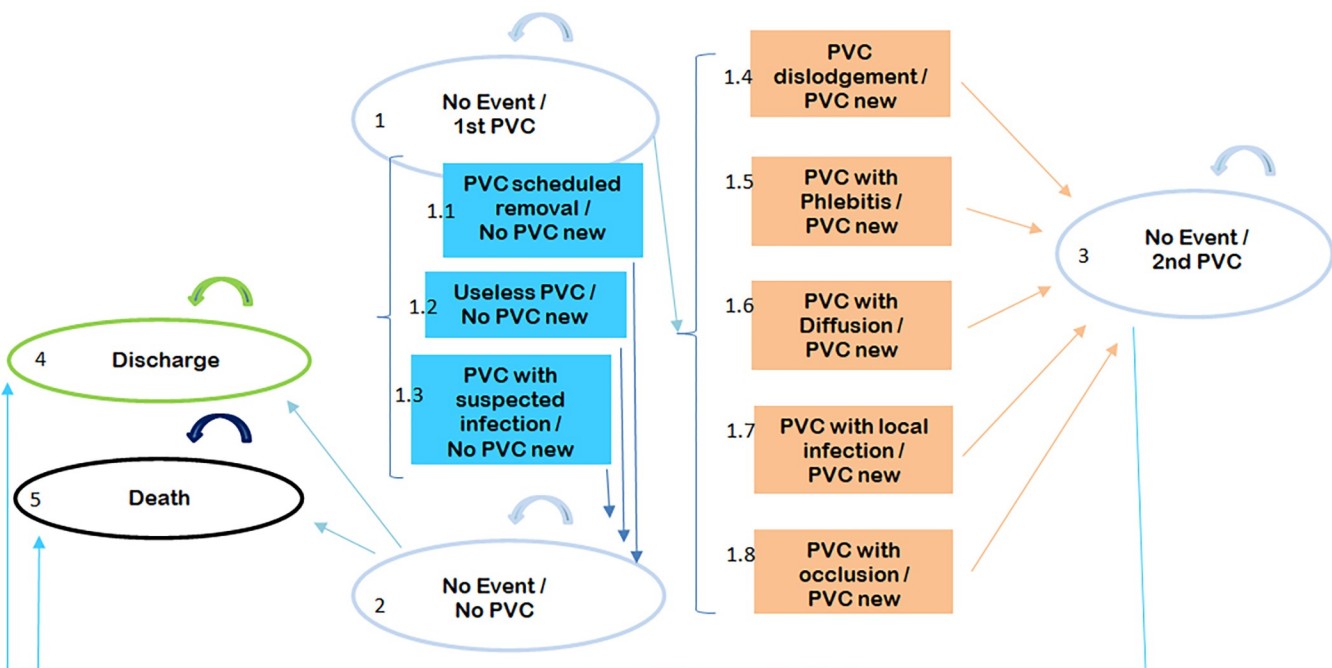

**Fig 1. Observed model structure from CLEAN-3 database (BDs strategy, SDs strategy)–Markov diagram.** BDs: Bundle of devices, SDs: Standard devices, PVC: Peripheral Venous Catheter.

Instead of parametric Monte Carlo simulation, the *msm package* [10] allows to quantify uncertainty with nonparametric bootstrap methods for probabilistic sensitivity analysis and 95% confidence intervals (CI) calculations. To populate the model, data are specified as a series of observations, grouped by patient and sorted by increasing observational time from the patient entry in ED. At minimum there should be a data frame with variables indicating:

- The time of the observation,
- The observed state of the process,
- The subject identification number (ID).

Then all the observations are assumed to be from the same subject. The subject ID does not need to be numeric, but data must be grouped by subject, and observations must be ordered by time within subjects.

Figure below shows how a non-homogeneous semi-Markov model can work (Fig 2).

## Main assumptions

1. The cost of an event is independent from the outcome (survival or death or discharge): Statistical unit is the "global" patient. "Global" patient indicates a patient who could, during the hospital stay, be alive in the hospital, be alive and discharged of the hospital, or be died;

2. The estimated cost per event at the University Hospital of Poitiers is estimated for each compared intervention;

3. Discharge and Death Markov states are considered as absorbing states, e.g., patients cannot move from these states. As a consequence, an absorbing state is frequently valued at zero

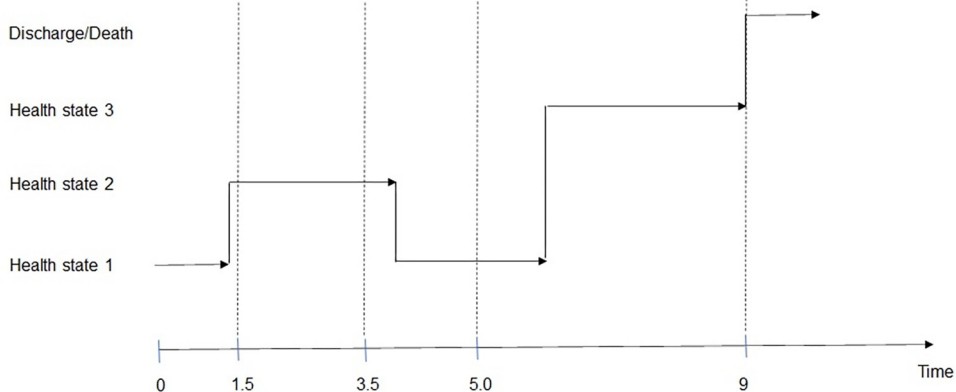

**Fig 2. Evolution of a multi-state model.** As an example case study within the msm package, the process is observed, for instance, on four occasions (source: msm package [10]).

cost, except here considering the step of catheter removal before discharge or death, if a catheter is in place.

4. As no data were available regarding 2<sup>nd</sup> catheter follow-up (date of removal), we considered the mean duration of the 2<sup>nd</sup> catheter was the difference in days between the censured time horizon (14 days) and the date of second catheter placement.

## Base case input parameters for the cost analysis

The base case analysis of the cost-effectiveness study has to be the most conservative case as possible, and the most representative case of real life, taking into account current hospital settings in France, and according to clinical experts, literature and RCTs. The patients' characteristics in the CLEAN-3 database were used for modeling (frequency and type of complications leading to catheter replacement, time of occurrence of complications, according to the two study groups).

The average cost of catheter insertion, replacement and removal, the average cost of each day of use of the venous line (excluding the cost of treatments administered) and the average cost of treating complications were estimated from the cost of necessary material and nursing time.

As the antiseptic cost was slightly different between the "chlorhexidine" and "povidone-iodine" solution, an average cost weighted for the number of patients included in CLEAN-3 study was used. Resources and unit costs were estimated following observations on practices at the University Hospital of Poitiers. Detailed resources use, nursing time, and unit costs are reported in the S1 File. For each patient group, the mean cost per patient was partly based on

**Table 2. Number of catheters per patient—Statistical unit: The global patient with catheterization (alive, discharge or dead).**

|  | Total Patients | SDs group | BDs group |
| --- | --- | --- | --- |
| Number of patients | 989 | 495 (50.1%) | 494 (49.9%) |
| Number of catheters | 1,39 | 730 (52.3%) | 666 (47.7%) |
| Number of catheters per patient | 1.40 | 1.50 | 1.30 |

SDs: Standard devices; BDs: Bundle of innovative devices.

**Table 3. Input parameters considered in the cost analysis–For 1 patient-catheter (Euro 2022).**

| | SDs group[*] | BDs group[*] |
|---|---|---|
| Unit cost: Placement initial catheter | 8.20 | 9.74 |
| Unit cost: Initial catheter removal or replacement | 2.32 | 2.26 |
| Unit cost: Placement second catheter | 8.45 | 9.98 |
| Total cost: Treatment for Dislodgement | 5.49 | 5.49 |
| Total cost: Treatment for Phlebitis | 12.27 | 12.27 |
| Total cost: Treatment for Diffusion | 4.09 | 4.09 |
| Total cost: Treatment for Local infection | 12.27 | 12.27 |
| Total cost: Treatment for Occlusion | 3.67 | 3.67 |
| Unit cost for 24 hours: Daily use of catheter | 12.31 | 13.27 |

SDs: Standard devices; BDs: Bundle of innovative devices.

[*]Source: University Hospital of Poitiers.

the mean number of catheters per patient (Table 2) and input parameters considered in the cost analysis (Table 3).

**Cost items for Markov states and transitional events.** Cost items for Markov states and transitional events are shown in Table 4.

**Costs per Markov states and transitional events (base case scenario).** Costs per Markov states and transitional events are shown in Table 5.

**Table 4. Costs items for Markov states and transitional events.**

| Main costs | Source/Data Provider | 1. No Events[*]/ 1st PVC | 1.1 E1 | 1.2 E2 | 1.3 E3 | 2. No Events[*]/ No PVC | 1.4 E4 | 1.5 E5 | 1.6 E6 | 1.7 E7 | 1.8 E8 | 3. No Events[*]/ 2nd PVC | 4. Discharge | 5. Death |
|---|---|---|---|---|---|---|---|---|---|---|---|---|---|---|
| Unit cost: Placement initial catheter | CHU Poitiers | X | | | | | | | | | | | | |
| Unit cost: Initial catheter removal | UHP | | X | X | X | | X | X | X | X | X | | | |
| Unit cost: Placement second catheter | UHP | | | | | | X | X | X | X | X | | | |
| Unit cost: Second catheter removal | UHP | | | | | | | | | | | | X | X |
| Total cost: Treatment for Dislodgement | UHP | | | | | | X | | | | | | | |
| Total cost: Treatment for Phlebitis | UHP | | | | | | | X | | | | | | |
| Total cost: Treatment for Diffusion | UHP | | | | | | | | X | | | | | |
| Total cost: Treatment for local infection | UHP | | | | | | | | | X | | | | |
| Total cost: Treatment for Occlusion | UHP | | | | | | | | | | X | | | |
| Unit cost for 24h: Daily use of catheter | UHP | X | | | | | | | | | | X[**] | | |

UHP: University Hospital of Poitiers; PVC: Peripheral venous catheter.

[*]Transitional events: E1 scheduled PVC removal; E2 useless PVC; E3 suspected infection (without event); E4 Dislodgement; E5 phlebitis; E6 diffusion; E7 local infection; E8 occlusion.

[**]As no data were available regarding 2nd catheter follow-up (date of removal), we considered the mean duration of the 2nd catheter was the difference in days between the censured time horizon (14 days) and the date of second catheter placement.

**Table 5. Costs per Markov states and transitional events (Euro 2022).**

| Markov State Event[1] | Costs for 1 catheter SDs group | Costs for 1 catheter BDs group | Costs for 1 patient SDs group | Costs for 1 patient BDs group |
|---|---|---|---|---|
| 1. No Events*/ 1st PVC | 8.20+12.31 = 20.51 (1er jour) 12.31n**(>J2) | 9.74+13.27 = 23.01 (1er jour) 13.27n**(>J2) | 20.51x1.5 [a] (1er jour) = 30.76 12.31 x1.5[a] = 18.47 (>J2) | 23.01x1.3 [b] (1er jour) = 29.91 13.27 x1.3[b] = 17.25 (>J2) |
| 1.1 E1 | 2.32 | 2.26 | 3.47 | 2.94 |
| 1.2 E2 | 2.32 | 2.26 | 3.47 | 2.94 |
| 1.3 E3 | 2.32 | 2.26 | 3.47 | 2.94 |
| 2. No Events*/ No PVC | 0.00 | 0.00 | 0.00 | 0.00 |
| 1.4 E4 | 26.77−8.20−2.32 = 16.25 | 29.73−9.74−2.26 = 17.73 | 24.38 | 23.05 |
| 1.5 E5 | 33.55−8.20−2.32 = 23.03 | 36.51−9.74−2.26 = 24.51 | 34.55 | 31.86 |
| 1.6 E6 | 25.37−8.20−2.32 = 14.85 | 28.33−9.74−2.26 = 16.33 | 22.28 | 21.23 |
| 1.7 E7 | 33.55−8.20−2.32 = 23.03 | 36.51−9.74−2.26 = 24.51 | 34.55 | 31.86 |
| 1.8 E8 | 24.95−8.20−2.32 = 14.43 | 27.90−9.74−2.26 = 15.91 | 21.64 | 20.68 |
| 3. No Events*/ 2nd PVC | 12.31x n days*** | 13.27x n days*** | 12.31x n days x1.5 | 13.27x n days x1.3 |
| 4. Discharge | 2.32**** | 2.26**** | 3.47**** | 2.94**** |
| 5. Death | 2.32**** | 2.26**** | 3.47**** | 2.94**** |

[1] From CLEAN-3 database.

PVC: Peripheral venous catheter.

* Events: E1, scheduled PVC removal; E2, useless PVC; E3, suspected infection; E4, Dislodgement; E5, phlebitis; E6, diffusion; E7, local infection; E8, occlusion.

** Unit cost for daily use of catheter x Number of catheter-days.

*** As no data were available regarding 2nd catheter follow-up (date of ablation), we considered the mean duration of the 2nd catheter was the difference in days between the censured time horizon (14 days) and the date of second catheter placement.

**** Absorbing states are generally valued at zero cost, except here if we consider the step of removing the catheter before discharge or death, for a patient in state 1 (1st PVC) or state 3 (2nd PVC).

[a] Number of catheters per patient in SDs group (control group).

[b] Number of catheters per patient in BDs group (experimental group).

**Potential additional hospital length of stay due to studied complications.** We can estimate the additional LOS due to studied complications based on observational data from CLEAN-3 database through the following calculation:

*Mean hospital LOS in patients with complications–Mean hospital LOS in patients without complications.*

A preliminary analysis showed that a mean difference in hospital LOS was of nearly +4 days. This original result will be studied by the University Hospital of Poitiers (UHP). Indeed, considering the studied complications could not clinically induce such an increase in LOS, UHP clinical experts and biostatisticians wish to study if other explanations could be found, as patient characteristics for instance.

Therefore, the working group validated the idea for evaluating the intrinsic effect of bundled catheters by considering only the costs related to catheters and studied complications (insertion, duration of use, removal, treatment of studied complications, replacement).

## Designing optimal cost-effectiveness model from observed CLEAN-3 individual patient data

**Influence of catheter strategy on prevention of unscheduled PVC removal due to complications.** As a reminder, among the patients with unscheduled PVC removal due to complications, there were 235 patients (47.5%) in the SDs group and 172 patients (34.8%) in the

BDs group. The difference in not-adjusted on $1^{st}$ catheter-time proportion was statistically significant (p = 0.00006). The result of the Fisher's exact test also indicates that the mean odds-ratio (OR) was of 1.69 (95%CI: 1.30; 2.20). Patients in the SDs group have between 1.3 and 2.2 times the risk of having an unscheduled catheter removal. Patients in the BDs group are therefore more protected on this criterion. We performed a logistic regression of the probability of being in the unscheduled PVC removal state as a function of the device group exposure. Based on this logistic model, the SDs group (p = 0.00006; 95%CI OR: 1.31; 2.18) was a statistically significant variable to explain unscheduled PVC removal. The duration of hospitalization was comparable between the BDs group and the SDs group, with durations of 10.4 and 9.3 days, respectively (p = 0.1902).

**Influence of duration of $1^{st}$ catheter exposure on unscheduled PVC removal due to complications.**    The above results did not consider the duration of $1^{st}$ catheter exposure in each group. Among the patients with unscheduled PVC removal due to complications, only the quantitative variable "Duration of $1^{st}$ catheterization in days" was statistically different between the two groups (Wilcoxon rank sum test with continuity correction: p = 0.004), with a longer duration of $1^{st}$ catheterization on average in the BDs group (2.12 versus 1.82 days). As a consequence, for taking into account the difference in $1^{st}$ catheter-time between the two groups, we performed a comparison per one $1^{st}$ PVC-day. The number of patients with unscheduled PVC removal per one $1^{st}$ PVC-day was of 16 patients (out of 100) in the BDs group and of 26 patients (out of 100) in the SDs group. A nonparametric bootstrap comparison has been carried out; the percentile method estimated a 95%CI of [0.15; 0.18], e.g., between 15 and 18 patients out of 100 in the BDs group and [0.24–0.28], e.g., between 24 and 28 patients out of 100 in the SDs group. This bootstrap sensitivity analysis estimated a mean difference in number of patients of -0.10 (95%CI: [-0.12; -0.07]), e.g., the BDs strategy prevented in mean 10 (95%CI: [7; 12]) patients (out of 100) per one $1^{st}$ PVC-day with unscheduled $1^{st}$ PVC removal due to complications.

**Influence of devices group on the proportion of patients discharged from the hospital (absorbing state) before the end of the study.**    Without adjustment for catheter time, the proportion of patients, with or without unscheduled PVC removal, discharged from the hospital before the end of the study was comparable (Fisher's Exact Test: p = 1) between the BDs (152 patients, 30.8%) and SDs groups (153 patients, 30.9%). With adjustment for catheter time, the proportion of patients discharged from hospital (per 1000 patient-hours of catheter) was comparable between the BDs (95%CI: [2.9; 6.6]) and SDs (95%CI: [3.8; 7.5]) groups. For patients with unscheduled PVC removal, after adjustment for catheter time, the proportion of patients discharged from the hospital (per 1000 patient-hours of catheter) were comparable between the BDs (95%CI: [0;1.2]) and SDs (95%CI: [0; 1.8]) groups. For patients without unscheduled PVC removal, after adjustment for catheter time, the proportions of patients discharged from the hospital (per 1000 patient-hours of catheter) were comparable between the BDs (95%CI: [2.4; 6.1]) and SDs (95%CI: [2.7; 6.5]) groups. This result validates the approach of not considering the cost of the hospital stay in our cost-effectiveness study, but rather focusing on the costs of unscheduled catheter removals, catheter replacements and treating complications.

**Influence of devices group on the proportion of patients in death state (absorbing state).**    Without adjustment for catheter time, the proportion of deceased patients was comparable (Fisher's Exact Test: p = 0.4206) between the BDs (8 patients, 1.6%) and SDs groups (5 patients, 1%). After adjustment for catheter time, the proportion of deceased patients (per 1000 patient-hours of catheter) were comparable between the BDs (95%CI: [0; 1.05]) and SDs (95%CI: [0; 0.96]) groups.

**Final design of the cost-effectiveness model based on the above "influence" observed results.** These results above were dependent on the devices group and the duration of catheter exposure. The analysis of the CLEAN-3 IPD showed that all unscheduled PVC removal events occurred through a maximum time horizon of 14 days, so we decided to calculate the intrinsic cost-effectiveness results for a 14-day time horizon in medical wards, taking into account all observed events due to 1st catheter complications in each devices group. Accordingly, the NH-SMC model was performed on this basis, taking into account a 14-day time horizon in hospital including ED visit (for base case analysis), and making assumption that all patients were catheterized during this common time period for each of the two devices groups. Also, we performed a Non-Homogeneous Semi-Markov Chain (NH-SMC) analysis on "Global" patient sample (considering observed results, as opposed to simulated results).

## Scenario analyses

**Scenario analysis 1.** A Non-Homogeneous Semi-Markov Chain analysis (observed results) on "Global" patient sample. Time of observation was of 2–14 1st PVC-days (considering catheterized patients with catheter duration more than 24 hours or less than 24 hours with a studied complication). This scenario allows the exclusion of catheters inserted for a short foreseeable period of time, for which the benefit of the bundle on the prevention of complications is low according to the results of CLEAN 3 [5].

**Scenario analysis 2.** A NH-SMC analysis (observed results) on "Global" patient sample: Time of observation was of 0–14 1st PVC-days without PVC complication costs. This scenario removes the cost of complications for which management is not standardized between hospitals, potentially favoring the bundle group.

**Sensitivity analyses.** The estimated transition probability matrix of the NH-SMC model was based on individual patient data from CLEAN-3 study [5]. This analysis can be considered as a non-homogeneous Semi-Markov Chain Monte Carlo modeling which takes into account time dependency of state transition, duration in each state, and individual path of states through time. Instead of parametric Monte Carlo simulation, the *msm* package [10] allows to quantify uncertainty with nonparametric bootstrap methods. Nonparametric bootstrap 95% confidence intervals (95%CI) were estimated for the transition probability matrix between health states, and the mean cost per patient for a 14-day time horizon in hospital (including ED visit). The bootstrap method that has been adopted was that of the *boot.ci* algorithm from the boot package [11–13]. This function generates 5 different types of equi-tailed two-sided nonparametric confidence intervals. These are the first order normal approximation, the basic bootstrap interval, the studentized bootstrap interval, the bootstrap percentile interval, and the adjusted bootstrap percentile (BCa) interval. We used the bootstrap percentile interval (*perc*) method.

## Results

## The observed "global" patient (alive, discharge or dead) from CLEAN-3 IPD: Base case scenario

**Effectiveness: Distribution of patients through health states and transitional events (observed Markov cycle: Each 1.4 days, a patient can change his/her health status) and duration in each non absorbing health state.** The NH-SMC modeling estimated that patients change health states, or remain in the same health state, every 1.4 days. This Markov cycle is thus close to the observable realities of the Clean-3 database. After 1.4 days of 1st catheterization, 58% of patients were still in health state 1 (no event, 1st PVC) in BDs group,

compared to 51% in SDs group. After 2.8 days of 1st catheterization, 27% of the patients were still in health state 1 (no event, 1st PVC in place) in BDs group, against 19% in SDs group. After 4.2 days of 1st catheterization, 11% of patients were still in health state 1 (no event, 1st PVC) in BDs group, compared to 6% in SDs group. The number of patients in health state 3 (no event, 2nd PVC in place), e.g., the Markov state that receives transitional events as dislodgement, phlebitis, diffusion, local infection or occlusion (the 1st PVC-related complications studied), was higher in the SDs group than in the BDs group (24% in the SDs group after 1.4 days of 1st catheterization, compared to 15% in the BDs group; 45% at 4.2 days compared to 30%).

The mean duration in health state 1 (no event, 1st PVC in place) was lower (1.82 days versus 2.12 days; p = 0.004) in SDs group compared with BDs group. The mean duration in health state 3 (no event, 2nd PVC in place; 7.79 versus 7.77 days) and in health state 2 (no event, no PVC in place) (3.59 versus 3.85 days) were comparable between patients in SDs and BDs groups, respectively.

**Cost-effectiveness results per patient.**   The adjustment coefficient for 1st catheter time in days per patient, for health state 1 (no event, 1st PVC in place) was of 1.1618 (2.1226 / 1.8269).

The cost-effectiveness results for the observed "global" patient (not simulated), for each of the two compared groups and for a 14-day time horizon in hospital are shown in Table 6.

The 95% confidence intervals for cost per patient did not overlap; the saving per patient was statistically significant at 0.05 level (between €32 and €54), preventing 12.65 patients (out of 100) with unscheduled PVC removal.

**Cost-effectiveness results per patient-PVC-day.**   The cost-effectiveness results for the observed "global" patient (not simulated), per patient-PVC-day, for each of the two compared groups and for a 14-day time horizon in hospital are shown in Table 7.

The 95% confidence intervals show that the saving per patient-PVC-day was estimated between €22 and €40, preventing 9.58 patients (out of 100) with unscheduled PVC removal per 1st PVC-day. As a consequence, BDs strategy is less costly and more effective than SDs strategy.

## Scenario analyses

**Scenario analysis 1: A non-homogeneous Semi-Markov Chain analysis on observed "Global" patient sample. Time of observation was of 2–14 1st PVC-days (considering catheterized patients with catheter duration more than 24 hours or less than 24 hours with a studied complication).**   *Cost-effectiveness results per patient.* The cost-effectiveness results for

**Table 6. Cost-effectiveness results per patient from observed data (CLEAN-3 database)–Observed global patient–Hospital-time Horizon: 14 days—Base case scenario.**

| Strategy | SDs Standard devices (Reference strategy) | BDs Bundled devices (Assessed strategy) |
|---|---|---|
| Mean cost per patient, adjusted on catheter-time (nonparametric bootstrap 95%CI) | €144.24 (€134.8; €154.2) | €102.11 (€95.4; €108.9) |
| Effectiveness: Number of patients with unscheduled 1st PVC removal (%) | 235/495 (47.47%) | 172/494 (34.82%) |
| Difference in Cost per patient (95%CI) | | €-42.13 (€-53.61; €-32.01) |
| Difference in Effectiveness | | -12.65 patients / 100 |
| ICER / Dominance | | Dominate SDs (less costly, more effective) |

CI: Confidence interval; PVC: Peripheral Venous Catheter; ICER: Incremental Cost-Effectiveness Ratio = Difference in Cost / Difference in Effectiveness.

**Table 7. Cost-effectiveness results per patient-PVC-day from observed data (CLEAN-3 database)–Observed global patient–Hospital-time Horizon: 14 days–Base case scenario.**

| Strategy | SDs Standard devices (Reference strategy) | BDs Bundled devices (Assessed strategy) |
|---|---|---|
| Mean cost per patient-1st PVC-day (95%CI) | €78.95 (€71.15; €87.86) | €48.10 (€43.85; €52.44) |
| Effectiveness: Number of patients with unscheduled PVC removal, per PVC-day (95%CI) | 0.2598 (0.2399; 0.2810) | 0.1640 (0.1524; 0.1766) |
| Difference in Cost per patient-PVC-day (95%CI) | | €-30.85 (€-39.64; €-22.41) |
| Difference in Effectiveness per patient-PVC-day (95%CI) | | -0.0958 (-0.1191; -0.0713) |
| ICER / Dominance | | Dominate SDs strategy (less costly, more effective) |

CI: Confidence interval; PVC: Peripheral Venous Catheter; ICER: Incremental Cost-Effectiveness Ratio = Difference in Cost / Difference in Effectiveness.

the observed "global" patient (not simulated), for each of the two compared groups are shown in Table 8.

The 95% confidence intervals for cost per patient did not overlap; the saving per patient was statistically significant at 0.05 level (between €40 and €65), preventing 12.65 patients (out of 100) with unscheduled PVC removal.

*Cost-effectiveness results per patient-PVC-day.* The cost-effectiveness results for the observed "global" patient (not simulated), per patient-PVC-day, for each of the two compared groups are shown in Table 9.

The 95% confidence intervals show that the saving per patient-PVC-day was estimated between €23 and €41, preventing 10.27 patients (out of 100) with unscheduled PVC removal per 1st PVC-day. As a consequence, BDs strategy is less costly and more effective than SDs strategy.

**Scenario analysis 2: A NH-SMC analysis on observed "Global" patient sample: Time of observation was of 0–14 1st PVC-days without 1st PVC complication costs.** *Cost-effectiveness results per patient.* The cost-effectiveness results for the observed "global" patient (not simulated), for each of the two compared groups are shown in Table 10.

The 95% confidence intervals for cost per patient did not overlap; the saving per patient was statistically significant at 0.05 level (between €28 and €48), preventing 12.65 patients (out

**Table 8. Cost-effectiveness results per patient from observed data (CLEAN-3 database)–Observed global patient–Scenario analysis 1.**

| Strategy | SDs Standard devices (Reference strategy) | BDs Bundled devices (Assessed strategy) |
|---|---|---|
| Mean cost per patient, adjusted on catheter-time (nonparametric bootstrap 95%CI) | €169.42 (€160; €179) | €117.87 (€110; €126) |
| Effectiveness: Number of patients with unscheduled PVC removal (%) | 235/495 (47.47%) | 172/494 (34.82%) |
| Difference in Cost per patient (95%CI) | | €-51.55 (€-64.57; €-40.15) |
| Difference in Effectiveness | | -12.65 patients / 100 |
| ICER / Dominance | | Dominate SDs (less costly, more effective) |

CI: Confidence interval; PVC: Peripheral Venous Catheter; ICER: Incremental Cost-Effectiveness Ratio = Difference in Cost / Difference in Effectiveness.

**Table 9. Cost-effectiveness results per patient-PVC-day from observed data (CLEAN-3 database)–Observed global patient–Scenario analysis 1.**

| Strategy | SDs Standard devices (Reference strategy) | BDs Bundled devices (Assessed strategy) |
|---|---|---|
| Mean cost per patient-1st PVC-day (95%CI) | €78.98 (€71.12; €87.68) | €47.32 (€42.96; €52.25) |
| Effectiveness: Number of patients with unscheduled PVC removal, per PVC-day (95%CI) | 0.27 (0.25; 0.29) | 0.17 (0.16; 0.18) |
| Difference in Cost per patient-PVC-day (95%CI) | | €-31.39 (€-41.16; €-23.27) |
| Difference in Effectiveness per patient-PVC-day (95%CI) | | -0.10 (-0.13; -0.08) |
| ICER / Dominance | | Dominate SDs strategy (less costly, more effective) |

CI: Confidence interval; PVC: Peripheral Venous Catheter; ICER: Incremental Cost-Effectiveness Ratio = Difference in Cost / Difference in Effectiveness.

of 100) with unscheduled PVC removal. As a consequence, BDs strategy is less costly and more effective than SDs strategy.

*Cost-effectiveness results per patient-PVC-day.* The cost-effectiveness results for the observed "global" patient (not simulated), per patient-PVC-day, for each of the two compared groups are shown in Table 11.

The 95% confidence intervals show that the saving per patient-PVC-day was estimated between €20 and €36, preventing 9.58 patients (out of 100) with unscheduled PVC removal per 1st PVC-day. As a consequence, BDs strategy is less costly and more effective than SDs strategy.

## Sensitivity analyses for base case scenario and scenario analyses

From the NH-SMC model, the bootstrap 95%CI lower and upper bounds of the three cost-effectiveness analyses were shown in Tables 6–11. For each of the base case scenario and scenario analysis, from a cost-effectiveness point of view, the BDs strategy statistically dominates the SDs strategy for the observed "global" patient because of the non-overlapping nature of 95%CI regarding the cost-effectiveness criteria. Indeed, the results of the probabilistic sensitivity analysis are illustrated on the cost-effectiveness (CE) plane (Fig 3), which describes the difference in number of unscheduled PVC removal per patient-1st PVC-day and the difference in

**Table 10. Cost-effectiveness results per patient from observed data (CLEAN-3 database)–Observed global patient–Scenario analysis 2.**

| Strategy | SDs Standard devices (Reference strategy) | BDs Bundled devices (Assessed strategy) |
|---|---|---|
| Mean cost per patient, adjusted on catheter-time (nonparametric bootstrap 95%CI) | €131.12 (€123.60; €139.30) | €94.19 (€88.60; €100.21) |
| Effectiveness: Number of patients with unscheduled PVC removal (%) | 235/495 (47.5%) | 172/494 (34.8%) |
| Difference in Cost per patient (95%CI) | | €-36.93 (€-47.8; €-27.9) |
| Difference in Effectiveness | | -12.65 patients / 100 |
| ICER / Dominance | | Dominate SDs (less costly, more effective) |

CI: Confidence interval; PVC: Peripheral Venous Catheter; ICER: Incremental Cost-Effectiveness Ratio = Difference in Cost / Difference in Effectiveness.

**Table 11. Cost-effectiveness results per patient-PVC-day from observed data (CLEAN-3 database)–Observed global patient–Scenario analysis 2.**

| Strategy | SDs Standard devices (Reference strategy) | BDs Bundled devices (Assessed strategy) |
|---|---|---|
| Mean cost per patient-1st PVC-day (95%CI) | €71.77 (€65.65; €79.66) | €44.37 (€40.52; €48.14) |
| Effectiveness: Number of patients with unscheduled PVC removal, per PVC-day (95%CI) | 0.26 (0.24; 0.28) | 0.16 (0.15; 0.18) |
| Difference in Cost per patient-PVC-day (95%CI) | | €-27.40 (€-36.44; €-20.19) |
| Difference in Effectiveness per patient-PVC-day (95%CI) | | -0.10 (-0.12; -0.07) |
| ICER / Dominance | | Dominate SDs strategy (less costly, more effective) |

CI: Confidence interval; PVC: Peripheral Venous Catheter; ICER: Incremental Cost-Effectiveness Ratio = Difference in Cost / Difference in Effectiveness.

cost per patient-1st PVC-day between the BDs strategy and the SDs strategy from 1,000 bootstrap replicates in each group. All the points of the CE plane (e.g., incremental cost-effectiveness ratios) from the simulations consistently confirmed that the BDs strategy is more effective and less costly than the SDs strategy.

## Discussion

Regardless of the scenario analysis performed, the health technology evaluated passed the cost-effectiveness test. Indeed, after adjustment for 1st catheter time and regarding results per patient for the base case scenario, the BDs strategy avoided 10 (95%CI: 7; 12) patients (out of 100) with unscheduled first catheter removal and the average cost saving induced by the BDs strategy was estimated at €42 (95%CI: €32; €54) per patient. Regarding the cost results per day of first catheter, the average costs per first catheter day were of €79 (95%CI: €71; €88) in the SDs group and of €48 (95%CI: €44; €52) in the BDs group. The average cost saving induced by the BDs strategy was estimated at €31 (95%CI: €22; €40) per patient-day of first catheter.

For the first scenario analysis (time of observation of 2–14 1st PVC-days, considering catheterized patients with catheter duration more than 24 hours or less than 24 hours with a studied complication), the BDs strategy avoided 10 (95%CI: 8; 13) patients (out of 100) with unscheduled first catheter removal and the average cost saving induced by the BDs strategy was estimated at €51 (95%CI: €40; €65) per patient. Regarding the cost results per day of first catheter, the average costs per first catheter day were of €79 (95%CI: €71; €88) in the SDs group and of €47 (95%CI: €43; €52) in the BDs group. The average cost saving induced by the BDs strategy was estimated at €31 (95%CI: €23; €41) per patient-day of first catheter.

For the second scenario analysis (time of observation of 0–14 1st PVC-days, without considering 1st PVC-related complications costs), the BDs strategy avoided 10 (95%CI: 7; 12) patients (out of 100) with unscheduled first catheter removal and the average costs per patient were respectively €131 (95%CI: €124; €139) in the SDs comparator group and €94 (95%CI: €89; €100) in the BDs group. The average cost saving induced by the BDs strategy was estimated at €37 (95%CI: €28; €48) per patient. Regarding the cost results per day of first catheter, the average costs per first catheter day were of €72 (95%CI: €66; €80) in the SDs group and €44 (95% CI: €41; €48) in the BDs group. The average cost saving induced by the BDs strategy was estimated at €27 (95%CI: €20; €36) per patient-day of first catheter.

In Clean 3, 407 patients (41%) required an unscheduled PVC removal before discharge. Insertion of a second catheter in connection with unscheduled removal of the first catheter

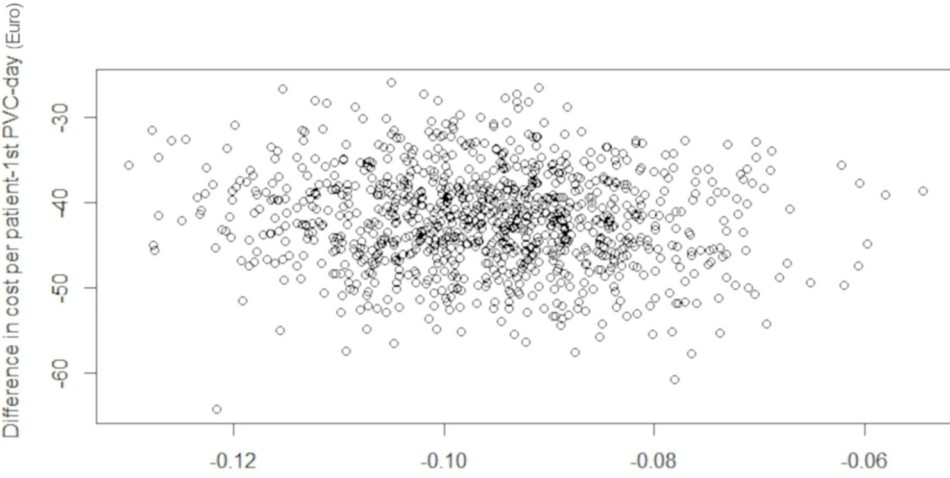

**Fig 3. Probabilistic sensitivity analysis: Cost-effectiveness plane for the base case analysis.** PVC: Peripheral Venous Catheter.

was more frequent (p = 6.058e-5) in the standard approach group (235 patients [47.5%]) than the bundled approach group (172 patients [34.8%]). We note that the mean catheter-time was different between the two groups; they were longer in the group of patients with bundled approach. The results of statistical tests indicate that it was necessary to adjust the comparisons between the two devices groups on 1[st] catheter-time.

After adjusting for 1[st] catheter-time, the rates of patients discharged from hospital (per 1000 patient-hours of catheter), before the end of the study and not having had an unscheduled catheter removal, were statistically comparable between the bundled approach and the standard approach groups (95% confidence intervals overlapped). We conclude that the "hospital discharge" condition has not influenced the differential cost-effectiveness outcome between patients in the bundled approach group and those in the standard approach group. This result validates the approach of not considering the cost of the hospital stay in our cost-effectiveness study, but rather focusing on the costs of complications and unscheduled catheter removals. Similarly, the rates of patient deaths (per 1000 patient-hours of catheter time) were comparable between the bundled approach and the standard approach groups.

The non-homogeneous multi-state semi-markovian model (NH-SMC) in continuous time is a suitable mathematical tool to be fitted to longitudinal data based on individual patient data (IPD) available in CLEAN-3 database (Poitiers University Hospital). The literature in this field frequently offers examples based on static decision tree models, used for both cost-effectiveness or cost-benefit studies [14–16], except for the latest Maunoury et al. paper on this topic [17]. The feature of the current modeling relates to the fact that it is based on real-life individual patient data, and not on published mean values from the literature. The time-dependence addressed here (e.g., evolution of the risk of developing a PVC-complication with increased catheterization time) corroborates that the non-homogeneous modeling approach is suitable considering the nature of the available data and the medical wards settings.

The rationale of the sensitivity analysis for scenario 1 (Hospital-Time Horizon: 2–14 days for all patients and 0–1 day for patients with complications) was to assess the intrinsic effect of the health technology studied by allowing it time to act beyond the first 24 hours of catheter placement, as reported in the manuscript of CLEAN3 [5]. The rationale of the sensitivity

analysis for scenario 2 (not taking into account the costs of complications leading to unscheduled PVC removal for the first catheter) was to evaluate the economic impact of the technology studied by considering only the costs linked to the use of catheters, without considering the other events that the patient may experience during the studied time horizon in medical wards.

Few studies have investigated costs associated with PVC insertion, daily maintenance, and catheter removal. These costs vary according to the organization of care, the type of device used, the year the study was conducted. They should include the cost of waste disposal, which is rarely the case. In addition, due to inflation, these costs need to be adjusted to allow for comparison between studies conducted at different times.

In the Netherlands, the average costs for catheter insertion were estimated to be €11.67 in 2019, but varied significantly with the number of cannulation attempts, from €9.32 for patients with a successful first attempt to €65.34 when five attempts were required [18]. In Australia, the average cost in 2016 was €6.39 (€6.75 at 2019 costs, the year CLEAN-3 was performed) [19]. Costs of up to about €30 for the first attempt have been reported by others [20]. Through a series of observations of caregiver practices in our hospital, we estimated similar costs (€8.20 with the standard approach and €9.74 with the bundle approach). The cost of catheter removal was estimated at €1.92 (€1.95 at 2019 costs) in an Australian study [19], a value close to our study estimate (€2.32 with the standard approach and €2.26 with the bundled approach).

Catheter failure is a frequent event, occurring in 41% of CLEAN-3 patients, a figure close to the average of 45% observed in previous randomized trials [2]. Catheter failure increases costs because of the need to change the catheter and to handle the complication. Here again, few data are available in the literature concerning the cost of changing catheters or treating complications. Indeed, there is no protocol for the management of these complications applied universally. We therefore estimated them on the basis of series of observations in our hospital. Average costs to handle one episode of diffusion (€4.09 vs [€0.12-€19.41]), occlusion (€3.67 vs [€0.25-€14.03]) and phlebitis (€12.27 vs [€0.04-€14.49]) were in the range of values reported in China [21].

This NH-SMC model has some limitations. First, it was built on a single clinical study because it was the only RCT available with this particular product. Second, the cost-effectiveness analysis was based on a scenario specific to French medical wards, with their own protocol to treat complications. As a consequence, the NH-SMC model cannot be directly transposed to other settings or other countries with different settings. This transposition would require local individual data on time-dependent probabilities of transition among health states at the daily level. Further studies involving other countries are needed to generalize our results and therefore our findings do not necessarily predict similar cost effectiveness of bundle of the studied devices in other countries or in specific patients' subgroups. Nevertheless, our results show that 1) globally our costs for PVC insertion, daily use and ablation are in the low range of the literature data and 2) excluding the costs of complications which could be a questionable point (cf. Scenario analysis 2), the BDs strategy keep a significant benefit. This study also has the non-technical limitation of being sponsored by industry (the BD Company). However, an external research organization (Statesia) was hired by the University Hospital of Poitiers to handle independently the development of the cost-effectiveness model and the data analysis to remove any possible bias. Non-BD authors have worked for the preparation of the manuscript, with the final version being approved by all non-BD authors prior to submission.

## Conclusion

According to the sensitivity analysis (nonparametric bootstrap 95% confidence intervals) which addresses the level of uncertainty of the mean results, and the results highlighted in this

study, the bundled devices (BDs) passed the test for cost-effectiveness within a conservative scenario defined through the base case scenario. The BDs strategy is significantly more effective to prevent PVC-complications and, as a consequence, unscheduled PVC-removal, when compared to the standard devices approach (SDs), with significant savings for the hospital. As a consequence, from a cost-effectiveness point of view, we can recommend the routine use of these bundled devices for patients in medical wards.

## Supporting information

**S1 File. Detailed resources and costs of the cost-effectiveness study.** CHG: Chlorhexidine Gluconate gel; PVI: povidone iodine-alcohol solution; PVC: Peripheral Venous Catheter. (XLSX)

## Acknowledgments

The authors would like to thank all stakeholders involved in the CLEAN-3 Study, which is at the origin of this cost-effectiveness study, including the members of the CLEAN-3 study group: Emergency Department and SAMU 86 Centre 15, University Hospital of Poitiers, France (J Guenezan MD, N Marjanovic MD, B Drugeon MD, R O Neill MD, Prof O Mimoz PhD); University of Poitiers, UFR de Médecine-Pharmacie, Poitiers, France (J Guenezan MD, N Marjanovic MD, Prof F Roblot PhD, P Palazzo PhD, M Pichon PhD, M Boisson PhD, Prof D Frasca PhD, Prof O Mimoz PhD); INSERM U1070, Pharmacology of Anti-Infective Agents, Poitiers, France (J Guenezan MD, Prof F Roblot PhD, M Pichon PhD, M Boisson PhD, Prof O Mimoz PhD); Geriatric Department, University Hospital of Poitiers, France (E Liuu MD); Department of Infectious and Tropical Diseases, University Hospital of Poitiers, France (Prof F Roblot PhD); Neurology Department, University Hospital of Poitiers, France (P Palazzo PhD); Pneumology Department, University Hospital of Poitiers, France (V Bironneau MD); Hepato-Gastro Enterology Department, University Hospital of Poitiers, France (F Prevost MD), Methodology-Data-Management platform, University Hospital of Poitiers, France (J Paul MSc); Department of Infectious Agents, Bacteriology-Hygiene Laboratory, University Hospital of Poitiers, France (M Pichon PhD); Department of Anesthesia, Intensive Care and Peri-Operative Medicine University Hospital of Poitiers, France (M Boisson PhD, Prof D Frasca PhD); INSERM U1246, Methods in Patients-centered outcomes and Health Research–SPHERE, Nantes, France (Prof D Frasca PhD).

## Author Contributions

**Conceptualization:** Franck Maunoury, Bertrand Drugeon, Olivier Mimoz, Jeremy Guenezan.

**Data curation:** Bertrand Drugeon, Olivier Mimoz.

**Formal analysis:** Franck Maunoury.

**Funding acquisition:** Olivier Mimoz.

**Investigation:** Bertrand Drugeon, Matthieu Boisson, Nicolas Marjanovic, Raphael Couvreur, Olivier Mimoz, Jeremy Guenezan.

**Methodology:** Franck Maunoury.

**Project administration:** Franck Maunoury.

**Supervision:** Franck Maunoury, Olivier Mimoz.

**Validation:** Olivier Mimoz.

**Visualization:** Franck Maunoury.

**Writing – original draft:** Franck Maunoury, Bertrand Drugeon.

**Writing – review & editing:** Franck Maunoury, Bertrand Drugeon, Matthieu Boisson, Nicolas Marjanovic, Raphael Couvreur, Olivier Mimoz, Jeremy Guenezan.

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
