## [Decision Letter · Decision Letter 0]

27 Apr 2022

PONE-D-22-07543Cost-effectiveness analysis of bundled innovative devices versus standard approach in the prevention of unscheduled peripheral venous catheters removal due to complications in FrancePLOS ONE

Dear Dr. Maunoury,

Thank you for submitting your manuscript to PLOS ONE. After careful consideration, we feel that it has merit but does not fully meet PLOS ONE’s publication criteria as it currently stands. Therefore, we invite you to submit a revised version of the manuscript that addresses the points raised during the review process.

The reviewers have commented on your above paper. They have suggested that this manuscript be revised according to the reviewers suggestions and resubmitted.  Provided you address the changes recommended, the manuscript will be accepted for publication

We look forward to receiving your revised manuscript.

Kind regards,

Prof. Raffaele Serra, M.D., Ph.D

Academic Editor

PLOS ONE

Journal Requirements:

"OM received funding for congress attendance, and research funding from Becton Dickinson. MB received personal fees from Becton Dickinson. All other investigators and authors declare no competing interests."

Additional Editor Comments:

The manuscript is interesting. There are only minor revision in order to have the manuscript ready for publication.

Reviewers' comments:

Reviewer's Responses to Questions

**Comments to the Author**

1. Is the manuscript technically sound, and do the data support the conclusions?

Reviewer #1: Partly

Reviewer #2: Yes

2. Has the statistical analysis been performed appropriately and rigorously? 

Reviewer #1: Yes

Reviewer #2: Yes

3. Have the authors made all data underlying the findings in their manuscript fully available?

Reviewer #1: Yes

Reviewer #2: Yes

4. Is the manuscript presented in an intelligible fashion and written in standard English?

Reviewer #1: Yes

Reviewer #2: Yes

5. Review Comments to the Author

Reviewer #1: The authors aimed to perform a cost-effectiveness analysis of bundled

devices (BDs) versus standard devices (SDs) for the prevention of unscheduled

peripheral venous catheter (PVC) removal due to complication from a French

investigator-initiated, open-label, single center, randomized-controlled, two-by-two

factorial trial (CLEAN-3 study).

The study is very interesting but I think that in the discussion section the role of biofilm should be discussed. For this purpose please cite and comment the following work: doi: 10.2174/1574887114666191018144739.

Reviewer #2: The article is interesting and well-written. Moreover, the statistical analysis il well-conducted. I have only the following minor comments:

1. In cost-effectiveness (CE) analysis (CEA), the CE plane is an important tool. It aims to clearly illustrate differences in costs and effects between different strategies (in your case, bundled devices, BDs, versus standard devices, SDs). The authors should add the CE plane of their analysis.

2. More details could be given about the estimated transition matrix.

3. An important component to any CE analysis is to assess whether the model is appropriate for the phenomena being examined, which is the purpose of model validation. In fact, very careful attention must be paid to the verification of a fundamental assumption which is the Markov property. Please discuss this aspect and perform model validation.

4. Line 91 - instead of “Statistical analyzes of observed” perhaps it is better to write “Statistical analysis of observed”.

5. Line 94 - The sentence about RStudio should be improved. To be more precise: “RStudio is an Integrated Development Environment for R, a programming language for statistical computing and graphics.”

6. Line 115: revise “by by”.

7. Always use the same number of decimals; see, just as an example, the p-values in the text.

8. Line 367: revise “Sensitivity analyzes”.

6. PLOS authors have the option to publish the peer review history of their article (what does this mean?). If published, this will include your full peer review and any attached files.

Reviewer #1: No

Reviewer #2: No

---

## [Author Response · Author response to Decision Letter 0]

4 May 2022

Response to PLOS One review

PLOS ONE Decision:

PLOS ONE <em@editorialmanager.com>

 27 avr. 2022 16:54 

À Franck Maunoury <franck.maunoury@statesia.com>

PONE-D-22-07543

Cost-effectiveness analysis of bundled innovative devices versus standard approach in the prevention of unscheduled peripheral venous catheters removal due to complications in France

PLOS ONE

Dear Dr. Maunoury,

Thank you for submitting your manuscript to PLOS ONE. After careful consideration, we feel that it has merit but does not fully meet PLOS ONE’s publication criteria as it currently stands. Therefore, we invite you to submit a revised version of the manuscript that addresses the points raised during the review process.

The reviewers have commented on your above paper. They have suggested that this manuscript be revised according to the reviewers suggestions and resubmitted. Provided you address the changes recommended, the manuscript will be accepted for publication

We look forward to receiving your revised manuscript.

Kind regards,

Prof. Raffaele Serra, M.D., Ph.D

Academic Editor

PLOS ONE

Response to PLOS One review –May 4 2022

Dear Prof. Raffaele Serra,

On behalf of my co-authors, I would like to thank you and the reviewers for your constructive feedback on our manuscript number PONE-D-22-07543. Please find enclosed our rebuttal letter in response to each point brought up by the academic editor and the reviewers.

We look forward to hearing from you in due time regarding our submission and to respond to any further questions and comments you may have.

Sincerely yours,

Dr. Franck Maunoury

Response to comments from Academic Editor, PLOS One (Prof. Raffaele Serra)

Journal Requirements:

Response: We have ensured that our manuscript meets the PLOS ONE’s style requirements.

"OM received funding for congress attendance, and research funding from Becton Dickinson. MB received personal fees from Becton Dickinson. All other investigators and authors declare no competing interests."

Response: "OM & JG received personal fees from Becton Dickinson, funding for congress attendance, and research funding from Becton Dickinson. MB received personal fees from Becton Dickinson. All other investigators and authors declare no competing interests."

Response: “This does not alter our adherence to PLOS ONE policies on sharing data and materials.” 

Response: Done. The updated Competing Interests statement is in our cover letter.

Response: We reviewed the reference list; it is complete and correct.

Additional Editor Comments:

The manuscript is interesting. There are only minor revision in order to have the manuscript ready for publication.

Response to Reviewers' comments:

Reviewer's Responses to Questions

Comments to the Author

1. Is the manuscript technically sound, and do the data support the conclusions?

Reviewer #1: Partly Response: See our responses to Reviewer #1

Reviewer #2: Yes 

2. Has the statistical analysis been performed appropriately and rigorously?

Reviewer #1: Yes

Reviewer #2: Yes

3. Have the authors made all data underlying the findings in their manuscript fully available?

Reviewer #1: Yes

Reviewer #2: Yes

4. Is the manuscript presented in an intelligible fashion and written in standard English?

Reviewer #1: Yes

Reviewer #2: Yes

5. Review Comments to the Author

Reviewer #1: The authors aimed to perform a cost-effectiveness analysis of bundled

devices (BDs) versus standard devices (SDs) for the prevention of unscheduled

peripheral venous catheter (PVC) removal due to complication from a French

investigator-initiated, open-label, single center, randomized-controlled, two-by-two

factorial trial (CLEAN-3 study).

The study is very interesting but I think that in the discussion section the role of biofilm should be discussed. For this purpose please cite and comment the following work: doi: 10.2174/1574887114666191018144739.

Response: Thank you very much for your general appreciation of our work and for this comment. In our study, we were mainly interested in non-infectious complications leading to catheter failure and replacement. Biofilm is an essential element leading to catheter-related infections. The role of biofilm in non-infectious catheter complications is less well documented. Therefore, we are not sure how to include this reference in our article. But we remain at the disposal of the reviewer if necessary.

Reviewer #2: The article is interesting and well-written. Moreover, the statistical analysis il well-conducted. I have only the following minor comments:

1. In cost-effectiveness (CE) analysis (CEA), the CE plane is an important tool. It aims to clearly illustrate differences in costs and effects between different strategies (in your case, bundled devices, BDs, versus standard devices, SDs). The authors should add the CE plane of their analysis.

Response: Thank you for your suggestion. The CE plane (Fig 3) has been added in the revised manuscript.

2. More details could be given about the estimated transition matrix.

Response: The estimated transition probability matrix is based on individual patient data from CLEAN-3 database. This analysis is considered as a non-homogeneous Semi-Markov Chain (NH-SMC) analysis which takes into account time dependency of health state transition, duration in each health state, and individual path of health states through time. The observed transitions among health states are shown on the Markov diagram (Fig 1).

The multi-state Markov model is a useful way of describing a process in which an individual moves through a series of states in continuous time. The msm package for R [ ] allows a general multi-state model to be fitted to longitudinal data. Data often consist of observations of the process at arbitrary times, so that the exact times when the state changes are unobserved. Kalbfleisch and Lawless[ ] and later Kay [ ] described a general method for evaluating the likelihood for a general multi-state model in continuous time, applicable to any form of transition matrix. The available information is the observed state at a set of times.

The next state to which the individual moves, and the time of the change, are governed by a set of transition intensities qrs(t; z(t)) for each pair of states r and s. The intensities may also depend on the time of the process t, or more generally a set of individual-specific or time-varying explanatory variables z(t). The intensity represents the instantaneous risk of moving from state r to state s:

qrs(t, z(t)) = lim P (S(t + δt) = s|S(t) = r)/δt (1)

δt→0

The intensities form a matrix Q whose rows sum to zero, so that the diagonal entries are defined by qrr = − ∑s≠r qrs.

To fit a multi-state model to data, we estimate this transition intensity matrix. Models whose intensities change with time are called time-inhomogeneous. We concentrate on Markov models here. The Markov assumption is that future evolution only depends on the current state. That is, qrs(t; z(t);Ft) is independent of Ft, the observation history Ft of the process up to the time preceding t. See Cox and Miller[ ] for a thorough introduction to the theory of continuous-time Markov chains. In a time-homogeneous continuous-time Markov model, a single period of occupancy (or sojourn time) in state r has an exponential distribution, with rate given by -qrr, (or mean -1/qrr). The remaining elements of the rth row of Q are proportional to the probabilities governing the next state after r to which the individual makes a transition. The probability that the individual’s next move from state r is to state s is -qrs/qrr. 

Transition probability matrix: 

The likelihood is calculated from the transition probability matrix P(t). For a time-homogeneous process, the (r; s) entry of P(t), prs(t), is the probability of being in state s at a time t+u in the future, given the state at time u is r. It does not say anything about the time of transition from r to s, indeed the process may have entered other states between times u and t+u. P(t) can be calculated by taking the matrix exponential of the scaled transition intensity matrix (Cox and Miller [iv]).

P (t) = Exp(tQ) (2)

In order to be as close as possible to observable realities, we have presented the cost-effectiveness results from the matrix of transition probabilities observed by the model (e.g., 'prevalence.msm' algorithm), and not from the matrix of transition probabilities estimated by the model. This approach is taken by the function prevalence.msm, which constructs a table of observed and expected numbers and percentages of individuals in each state at a set of times (see detailed response to #comment 3 below).

3. An important component to any CE analysis is to assess whether the model is appropriate for the phenomena being examined, which is the purpose of model validation. In fact, very careful attention must be paid to the verification of a fundamental assumption which is the Markov property. Please discuss this aspect and perform model validation.

Response: To compare the relative fit of two nested models, it is easy to compare their likelihoods. However it is not always easy to determine how well a fitted multistate model describes an irregularly-observed process. Ideally we would like to compare observed data with fitted or expected data under the model. If there were times at which all individuals were observed then the fit of the expected numbers in each state or prevalences can be assessed directly at those times. Otherwise, some approximations are necessary. We could assume that an individual’s state at an arbitrary time t was the same as the state at their previous observation time. This might be fairly accurate if observation times are close together. This approach is taken by the function prevalence.msm, which constructs a table of observed and expected numbers and percentages of individuals in each state at a set of times. A set of expected counts can be produced if the process begins at a common time for all individuals. Suppose at this time, each individual is in state 0. Then given n(t) individuals are under observation at time t, the expected number of individuals in state r at time t is n(t)P(t)0,r. If the covariates on which P(t) depends vary between individuals, then this can be averaged over the covariates observed in the data.

Comparing the observed and expected percentages in each state, we could see that the predicted number of individuals who die is under or over-estimated by the model. Such discrepancies could have many causes. One possibility is that the transition rates vary with the time since the beginning of the process; the age of the patient, or some other omitted covariate, so that the Markov model is non-homogeneous. This could be accounted for by modeling the intensity as a function of age, for example, such as a piecewise-constant function. The pci argument to msm can be used to automatically construct models with transition intensities which are piecewise-constant in time.

In this example, the hazard of death may increase with age, so that the model underestimates the number of deaths when forecasting far into the future. Another cause of poor model fit may sometimes be the failure of the Markov assumption. That is, the transition intensities may depend on the time spent in the current state (a semi-Markov process) or other characteristics of the process history. Accounting for the process history is difficult as the process is only observed through a series of snapshots. Semi-Markov models can be fitted to this type of data using phase-type distributions. Since version 1.4.1 the phase.states option to msm can be used to define some phase-type models.

As we wrote in the #comment 2 response, in order to be as close as possible to observable realities, we have presented the cost-effectiveness results from the matrix of transition probabilities observed by the model (e.g., 'prevalence.msm' algorithm), and not from the matrix of transition probabilities estimated by the model. As a consequence, the model (e.g., observed numbers and percentages of individuals in each state at a set of times) is implicitly appropriate for the phenomena being examined, which is indeed the demonstration of model validation. The Markov property has been discussed previously and implicitly handled by our non-homogeneous semi-Markov multi-state model (see explanations above).

4. Line 91 - instead of “Statistical analyzes of observed” perhaps it is better to write “Statistical analysis of observed”.

Response: Done.

5. Line 94 - The sentence about RStudio should be improved. To be more precise: “RStudio is an Integrated Development Environment for R, a programming language for statistical computing and graphics.”

Response: Done.

6. Line 115: revise “by by”.

Response: Done.

7. Always use the same number of decimals; see, just as an example, the p-values in the text.

Response: Done. We settled 2 decimals (if adapted), except for percentages (1 decimal), and for p-values which can be equal, for instance, to 0.01 or 0.00001 (we settled 5 decimals for the p-values considering all possible values, except for values equal to 0.001).

8. Line 367: revise “Sensitivity analyzes”.

Response: Done.

6. PLOS authors have the option to publish the peer review history of their article (what does this mean?). If published, this will include your full peer review and any attached files. 

Do you want your identity to be public for this peer review? For information about this choice, including consent withdrawal, please see our Privacy Policy.

Reviewer #1: No

Reviewer #2: No

 [i] Christopher Jackson. Multi-state modelling with R: the msm package. Journal of Statistical Software (2011) 38(8):1-29.

 [ii] J.D. Kalbfleisch and J.F. Lawless. The analysis of panel data under a Markov assumption. Journal of the American Statistical Association, 80(392):863–871, 1985.

 [iii] R.~Kay. A Markov model for analysing cancer markers and disease states in survival studies. Biometrics, 42:855–865, 1986.

 [iv] D.~R. Cox and H.~D. Miller. The Theory of Stochastic Processes. Chapman and Hall, London, 1965.

---

## [Editor Report · Decision Letter 1]

27 May 2022

Cost-effectiveness analysis of bundled innovative devices versus standard approach in the prevention of unscheduled peripheral venous catheters removal due to complications in France

PONE-D-22-07543R1

Dear Dr. Maunoury,

We’re pleased to inform you that your manuscript has been judged scientifically suitable for publication and will be formally accepted for publication once it meets all outstanding technical requirements.

Kind regards,

Prof. Raffaele Serra, M.D., Ph.D

Academic Editor

PLOS ONE

Additional Editor Comments (optional):

amended manuscript is acceptable
---

## [Editor Report · Acceptance letter]

2 Jun 2022

PONE-D-22-07543R1 

Cost-effectiveness analysis of bundled innovative devices versus standard approach in the prevention of unscheduled peripheral venous catheters removal due to complications in France 

Dear Dr. Maunoury:

I'm pleased to inform you that your manuscript has been deemed suitable for publication in PLOS ONE. Congratulations! Your manuscript is now with our production department. 

Kind regards, 

on behalf of

Prof. Raffaele Serra 

Academic Editor

PLOS ONE